# Study on the Relationship between Textile Microplastics Shedding and Fabric Structure

**DOI:** 10.3390/polym14235309

**Published:** 2022-12-05

**Authors:** Hong Cui, Changquan Xu

**Affiliations:** College of Textiles and Clothing, Yancheng Institute of Technology, Yancheng 224051, China

**Keywords:** microplastic, synthetic fiber, fabric structure, home washing, release

## Abstract

Microplastics refer to plastic fibers, particles or films less than 5 mm in diameter. Textile microplastics are an important form of microplastics, which can harm the ecological environment and human health. This paper studies the relationship between textile microplastic shedding and fabric structure to reduce microplastics pollution and reduce its impact on humans and the natural environment. Firstly, household washing is simulated by considering the main fabric type, the number of steel balls used in the washing, washing temperature, washing time and other influencing factors. An orthogonal test of the mixing level of the four factors is designed by selecting the fabric type, the number of steel balls used in washing, washing temperature and washing time, and the influencing factors is analyzed, and the best washing scheme is obtained. Then, under optimal washing conditions, the three factors and three levels of orthogonal test are designed to analyze the influence of fabric structure and external factors on the shedding of microplastics by changing the amounts of friction and insolation time. The results show that the microplastics released by knitted fabrics are significantly more under the same washing conditions than that of woven fabrics. Satin fabrics released the most microplastics and plain fabrics the least. In addition, among the external factors, the amount of friction significantly affects the production of microplastics.

## 1. Introduction

Microplastics are usually referred to as fragments, films, particles, and fibers with a diameter of less than 5 mm [1]. Microplastics can generally be divided into two levels: secondary microplastics and primary microplastics. Waste textiles, industrial products, shed fibers, garbage, and other small debris generated by physical and chemical means belong to secondary microplastics, while industrial raw materials belong to primary microplastics. Textile microplastics belong to secondary microplastics, which are a subclass of microplastics. Textile microplastics are mainly synthetic fibers and their products. The diameter of the synthetic fiber is very small, mostly at the micron or submicron level, and they are decomposed through physical and chemical reactions to produce microplastics [2]. The use of chemical fiber textiles in daily use and cleaning produce small microfibers on the surface of the fiber due to friction caused by mechanical mixing, and the small fibers between the fibers are also easy to drill out. The amount of microplastics produced by washing can reach hundreds of thousands or even millions of particles [3].

According to a new study published in the European Joint Gastroenterology Week, Philipp Schwabl et al. found for the first time that as many as nine different microplastics exist in the human body, indicating that microplastics have caused pollution to the human body [4]. Microplastic pollution is relatively common in rivers, lakes, and oceans. Microplastics have been detected in the Pearl River [5], Taihu Lake [6,7], Yangtze River [8], Wuhan urban river [9], etc. Microplastics have also been found in lakes on the Tibetan Plateau [10] and even in remote Antarctic waters [11]. Microplastics have even been found in fish, shrimp, crabs, and other aquatic organisms. This indicates that microplastic pollution in water has become widespread.

Microplastic pollution in the environment is closely related to the surrounding industrial and human activities. Microplastics are an emerging pollutant that accumulates large amounts in water and sediment samples collected from textile industrial areas. Browne et al. [12] found that the microplastics in the Yangtze River estuary near Shanghai had the highest fiber composition. In addition, Deng et al. [13] found that compared with the referenced agricultural area, the microplastic pollution levels in the industrial area were significantly higher. Wang et al. [14] showed that microplastics in informal landfills might leak into the surrounding environment without adequate protection. The density of microplastics is very small, and can be indirectly entered into organisms through water circulation and atmospheric circulation. Microplastics at the micron and nanometer levels can be exchanged at the cellular level, and toxic and harmful substances adsorbed by microplastics will cause harm to human body.

The harm caused by microplastics has three aspects. First, textile microplastics are mainly chemical fibers and their products. Due to the high molecular weight of some synthetic fibers and stable physical and chemical properties, they are difficult to be digested and metabolized by a microbial system, making it difficult to be degraded in the environment for dozens or even hundreds of years. Ingestion by the organism may cause physical damage to its organs, impair immune system function, and harm the growth and reproduction of the organism. Second, whether it is natural or synthetic fiber, to improve the performance and beauty of textiles to meet consumer needs, catalysts and dyes will inevitably be added during production, such as the molecular weight of organic heat stabilizers and other additives. A similar situation exists with other plastics. Most of these additives are low-to-medium molecular weight organic compounds or heavy metal inorganic compounds with high biological toxicity. When the textile is exposed to the natural environment to form fiber microplastics, the safety of organisms will be affected with the gradual release of the above chemicals [15]. Third, compared with larger-sized plastic waste, fiber microplastics have a larger specific surface area, and will enrich persistent pollutants and heavy metals from the surrounding environment during migration [16]. On the one hand, fibrous microplastics can become carriers of these pollutants and migrate through various ecosystems due to environmental factors such as wind and ocean currents, thereby expanding the range of pollutants. On the other hand, fiber microplastics can form toxic compounds through the adsorption of other pollutants, which have the toxicity of various other pollutants, cause physical and biochemical hazards to aquatic organisms, and therefore pose carcinogenic risks to humans [17].

Hartline et al. [18] carried out detergent-free washing experiments on five new jackets and five older jackets with front-mounted and top-mounted household roll-on washing machines. The study found that the mass of microfibers released by the top-mounted drum washing machine was about seven times that of the front-mounted washing machine. Older clothing that had been washed continuously for 24 h released more microfibers under the same washing regimen as the new clothing. At the same time, Pirc et al. [19] tested the microfiber produced by a new type of ultra-fine polyester fabric in the household washing process. They found that the release amount of fiber microplastics in the spinning process was about 3.5 times higher than that in the washing process. Napper et al. [20] studied the release of fibers in polyester, poly-cotton blends, and acrylic fabrics. It was found that poly-cotton fabrics consistently shed much less fiber than polyester or acrylic fabrics. Poly-cotton blends shed the least fiber without detergent and the most fiber when nonbiological detergent was used.

Researchers compared the release of microplastics from fabrics with different chemical compositions and the same weaving methods, such as polyester for plain weave, polyamide for plain weave, and acetate for twill weave (but other variables could not be fully controlled). Other variables must be controlled to determine the significance of a variable on different fabrics. By controlling for other variables, such as yarn and knitting method, Almroth et al. [21] studied the amount of microplastics released by fabrics with different chemical compositions and found that polyester/cotton fabrics always shed much less fiber than polyester or acrylic fabrics. The loose fabric structure sheds more, the worn fabric sheds more, and high-twist yarn is the preferred yarn to reduce the shedding.

Yang et al. [22] quantified microfiber shedding in three of the most common synthetic fabrics: polyester, polyamide, and acetate fabrics. It was found that the release of microfibers in the pulsing washing machine was greater than that in roller washing machine. Kelly et al. [23] developed a new method. By creating a calibration curve that quantifies the amount of fiber released by the textile during washing, it correlates with the quality of the fiber released. It was found that the water consumption and washing machine speed were the important factors affecting the release of fiber microplastics. Water consumption during washing is the biggest factor affecting the release of fiber microplastics. In addition, De Falco et al. [24] used a household washing machine to conduct an actual scale washing test on commercial garments. According to the properties and characteristics of laundry, the release of microfibers was analyzed. It was found that the yarn properties, such as the types of fibers that make up the yarn and their twist, affect the release of microfibers during washing. Cai et al. [25] studied that the release of microplastic fibers decreases with the reduction of repeated washing times for all kinds of textiles. After 5–6 times washing cycles, a small amount of fibers are continuously released, and the fiber length increases slightly.

Textile microplastics belong to a subclass of microplastics, which has great potential harm to the natural ecological environment and human health. The purpose of this paper is to study the impact of different fabric structures on textile microplastic shedding to reduce the production of textile microplastics and address the harm of microplastics to the environment. Different fabric structures mainly refer to the structures of different weaving methods and fabric structures. Therefore, this topic will be studied and analyzed in the following three aspects.

Firstly, the influence of fabric type and washing factor on the number of textile microplastics shed was studied. Factors such as fabric type, the number of washing balls used when washing, washing temperature and washing time, the four influencing factors of design for four factors mixed levels of the orthogonal test (fabric types for two levels, number of washing steel balls, washing temperature, and washing time for four levels), were analyzed in terms of their effects on the fiber loss quantity of washing to determine the optimal conditions. Then, the effects of fabric structure and external conditions on the amount of microplastics shed from textiles are studied. Three factors and three levels of orthogonal experiments are conducted for plain, twill, and satin fabrics after different insolation times and amounts of friction; the effects of fabric structure, insolation time and amount of friction on the number of shedded microplastics are analyzed. Finally, the shedded fiber microplastics are observed and analyzed. Throughout this research, the specific relationship between the shedding of textile microplastics and fabric structure will be obtained, thus providing a reference for how to reduce the shedding of textile microplastics.

## 2. Materials and Methods

### 2.1. Materials

This study uses polyester-knitted flat cloth, polyester woven fabric and plain, twill and satin fabric woven with the same ends and picks per inch and the same warp and weft yarn count by a semi-automatic punching machine are selected as the test materials. The characteristics of the tested fabrics are shown in Table 1.

The fabric samples are shown in Figure 1, Figure 2, Figure 3, Figure 4 and Figure 5.

The equipment used in this research is shown in Table 2.

### 2.2. Methodology

#### 2.2.1. Test Scheme for Fabric Types and Washing Conditions

In order to evaluate the influence of fabric type and washing conditions on the release of textile microplastics, the experiment is designed with a mixture level of multiple factors, including fabric type, the number of steel balls used in washing, washing temperature, and washing time. Steel balls refer to metal balls with a diameter of 7 mm, which increase mechanical force during washing. An orthogonal experiment is a design method that uses an orthogonal table to arrange and analyze multifactor experiments. It is tested by selecting some representative level combinations from all level combinations of test factors. We can understand the overall test situation by analyzing the test results and finding the optimal level combination. The orthogonal design refers to selecting representative test points from the comprehensive test points for testing. The orthogonal table has the following characteristics: First, each level of orthogonality in any column, occurs with an equal number of occurrences. All possible combinations of different levels between any two columns occur with equal frequency. Second, in terms of representativeness, all levels of any column appear so that part of the test includes all levels of all factors, and all level combinations of any two columns appear so that the test combination between any two factors is a comprehensive test. Third, in terms of comprehensive comparability, the number of occurrences of each level in any column is equal, and the number of occurrences of all level combinations between any two columns is equal so that the test conditions for each level of any factor are the same. This ensures that the interference of other factors is eliminated to the maximum extent in the effect of each column of factors at each level. Therefore, the influence of different levels of this factor on the test indexes can be comprehensively compared. In sum, the experiments arranged by the orthogonal table are characterized by balanced dispersion and orderliness. Equilibrium dispersion refers to the distribution of each factor level combination selected by the orthogonal table in all level combinations is uniform. These points are highly representative and can better reflect the overall test. Neat comparability means that each level of each factor is comparable. Because each level of each factor in the orthogonal table contains each level of the other factor in a balanced way, when comparing different levels of a factor, the effects of the other factors are offset by each other. Table 3 shows the level table of factors in the orthogonal test.

This experiment has four factors, among which factor A is at two levels and factors B, C and D are at four levels. Therefore, orthogonal table L_16_ (4^5^) is selected, as shown in Table 4. The vacant columns in the table are used to indicate error factors.

#### 2.2.2. Test Scheme for Fabric Structure and External Conditions

To analyze the influence of fabric structure and external conditions on the amount of textile microplastic shedding, a three-factor and three-level orthogonal test were designed in this experiment, including fabric structure, amount of friction and sunshine duration. The amount of friction refers to a multi-directional wear test in which the relative motion track between the sample made on the YG502 pilling tester and the wool standard abrasive is a circular track with a diameter of 40 mm. The wear mechanism of textile fiber is related to the friction coefficient of the fiber surface. The coefficient of the dynamic and static friction of textile fibers is different, and the coefficient of static friction is usually greater than that of dynamic friction. This difference is due to the “stick-slip” phenomenon in friction. The friction coefficient is affected by the relative slip speed of the fibers. The friction between fibers includes interface friction and lubrication friction [26,27]. Interfacial friction refers to the contact friction between solid fibers, which decreases with the increase of the slip speed. Lubricating friction refers to the friction of the viscous shear force of the fluid film between fibers, which increases with the increase of the slip speed [28,29]. Sunshine duration refers to the duration of exposure to sunlight. Table 5 is the level table of the orthogonal test factors.

This test contains three influencing factors, factor A, factor B and factor C are all at three levels, therefore orthogonal table L_9_ (3^4^) is selected, and is shown in Table 6.

## 3. Results and Discussions

### 3.1. Influence of Fabric Type and Washing Conditions on Microplastics Shedding

This orthogonal test was a four-factor orthogonal test with mixed levels (two levels for fabric type, four levels for washing temperature, washing time and a number of steel balls), and the microplastics mass (mg/10 g) produced per 10 g sample was taken as the result value. The test results and visual analysis table of this orthogonal experiment are completed according to the recorded test data, as shown in Table 7. K is the sum of the test results at the same level in the same column, such as K1 is the sum of the experiments at the 1 level in the same column, k1 is the average value of K1, and R is the range, which is the difference between the maximum and the minimum. The greater the range, the greater the degree of data dispersion, indicating that different levels of this factor have a greater impact on the results.

As can be seen from the data in Table 7, the order of range is A > B > D > C (fabric type > washing time > a number of steel balls > washing temperature), and factor A has the highest range. The range of factor B is second, indicating that the fabric type has the greatest influence on the release of microplastics, followed by the washing time, the number of steel balls, and the washing temperature has the least influence on the release of microplastics.

Under the same washing conditions, the amount of microplastics released from knitted fabrics is much higher than that from woven fabrics. The structure of the knitted fabric is coil and snare, which determines the elasticity and fluffiness of the knitted fabric. The interweaving of warp and weft yarns forms the structure of the woven fabric. Fabrics with fluffy structures are more likely to release fibers during washing than tight ones. The structural difference is why the release of microplastics from knitted fabrics is higher than that from woven fabrics.

To optimize the washing program, according to the size of the K value, the larger the value of K, the more textile microplastics are released. The optimized washing scheme is B4C2D4; when the washing time is 80 min, the washing temperature is 50 °C, the number of steel balls is 30, and the release of microplastics reaches the maximum. As can be seen from the data in Table 7, with the increase in washing time and the number of steel balls, the release of microplastics gradually increased. However, the release amount of microplastics did not change significantly with the washing temperature. With the increased washing temperature, the release amount of microplastics basically did not change significantly. This is consistent with the analysis result that washing temperature has the least effect on the release of textile microplastics in range analysis. Further analysis of variance is carried out for this orthogonal experiment, and the analysis results are shown in Table 8. The F-value in the table indicates the significant degree of influence of each factor on the index. The larger F is, the more significant the influence of factors on the index. The *p*-value is a measure of the difference. A *p*-value < 0.05 indicates that the factors significantly impact the release of microplastics. A *p*-value > 0.05 indicates that this factor has no significant effect on the release amount of microplastics. The smaller the *p*-value, the more significant the influence of the factors on the release of microplastics.

Analysis of variance is to determine the influence of controllable factors on the research results by analyzing and studying the contribution of variation from different sources to the total variation. Firstly, according to the ANOVA results in Table 8, the fabric type, washing time, and the number of steel balls significantly affect the release amount of microplastics. The fabric type has the largest F-value and the smallest *p*-value, indicating that among the above three factors, the fabric type has the most significant influence on the release amount of microplastics. The explanation of the reason is the same as the intuitive analysis. It is also caused by the loose coil structure of the knitted fabric and the close interwoven structure of the woven fabric. Secondly, the influence of washing time and the number of steel balls on the release amount of textile microplastics is also significant. Still, the F-value of washing time is larger, and the *p*-value is smaller, indicating that the influence of washing time is greater than that of the number of steel balls. The longer the washing time, the more microplastics are released. The influence of the number of steel balls is the size of the washing process of the fabric by mechanical force; the more the number of steel balls, the more the fabric in the washing by the mechanical blow, and the more the release of fabric microplastics. Finally, the same as the results of the visual analysis, washing temperature has no significant effect on the release amount of microplastics, indicating that the change of washing temperature almost does not affect the release amount of microplastics.

The mathematical software MATLAB is used to draw the three-dimensional surface map and contour map. The relationship between various factors and the release amount of textile microplastics is analyzed, as shown in Figure 6.

According to Figure 6, at the same washing time, the release amount of textile microplastics of the polyester knitted fabric is greater than that of the polyester woven fabric. The longer the washing time, the greater the amount of microplastics released. At the same washing temperature, the release of microplastics from polyester knitted fabric is greater than that from polyester woven fabric. With the increase in washing temperature, the release of microplastics first increased and then decreased. With the same number of washed steel balls, the release of microplastics from knitted polyester fabric is greater than that from woven polyester fabric. For the same type of fabric, the release of textile microplastics increased with the increased number steel balls in the washing container. In conclusion, the microplastic release amount of polyester knitted fabric is generally greater than that of polyester woven fabric. The reason is that knitted fabric has greater elasticity due to its coil and snare structure, and the short fibers on the surface of knitted fabric are easier to fall off during washing. However, woven fabric is made of interwoven warp and weft yarns, which have more interweaving points and tighter structure, and the short fibers on the surface are less to fall off during washing. The washing time and the number of steel balls are proportional to the release of microplastics, which is related to the time of washing on the fabric and the size of the external force. The longer the washing time, the greater the external force and the more microplastics are released. In addition, the influence of washing temperature on the release amount of microplastics shows that, within a certain range, the increase in temperature can increase the release amount of microplastics. However, increasing the temperature beyond a certain temperature decreases the release of microplastics.

### 3.2. Influence of Fabric Structure and External Conditions on Microplastics Shedding

This orthogonal test is a three-factor, three-level orthogonal test, and the microplastics mass (mg/10 g) produced per 10 g sample is taken as the result value. The test results and visual analysis table of this orthogonal experiment are completed according to the recorded test data, as shown in Table 9.

According to the data in Table 9, the order of range size is B > A > C (amount of friction > fabric structure > exposure time to sunlight). The range of factor B is the largest, followed by the range of factor A, indicating that the amounts of friction have the largest impact on the release of microplastics, fabric structure has the second-largest impact on the release of microplastics, and the sunshine time has the least impact on the release of textile microplastics.

When other conditions are the same, the more amount of friction, the more release of fabric microplastics. The woven fabric structure is compact, and the main damage form of fabric wear is fiber surface wear. At this time, the fiber segment in the yarn has very little mobility. Under the repeated friction between the fiber surface and the abrasive, the surface layer at both ends of the fiber and the buckling part appears fragmented and slightly broken, or a fibrillar structure. These fibrillar chips constantly fall off from the yarn, making the fiber fragile and easy to break. This kind of wear is mainly caused by debris loss due to the wear of the fiber surface, with a small amount of fiber fracture. The more amount of friction, the more fiber fragments formed, the greater the possibility of fiber fracture, and the more microplastics released during washing.

The order of microplastics released in the three fabric structures is satin > twill > plain under the same conditions. Satin weave has the least interlacing points among the basic fabrics. The fabric feels soft and has good elasticity. The surface of the satin fabric has an obvious warp and weft floating length. When under external force, the surface fibers are more likely to fall off. Twill fabrics have more interwoven warp and weft yarns than satin fabrics. The fabric feels soft and has good luster and elasticity. Due to more interlacing points and long floating thread, the release of twill fabric is less than that of satin fabric. The plain fabric has the most interlacing points, the cloth surface is flat and neat, and the microfiber shedding on the surface is the least when washing.

When other conditions are the same, the longer the sunshine time is, the more the fabric microplastics are released. Various environmental factors degrade the properties of fabrics during use. Light is one of the physical functions of aging. Photoaction is mainly caused by the thermal effects of photodegradation, photooxidation and photothermal conversion of fiber molecules in fabrics. The degradation and oxidation of fiber molecules change the degree of polymerization of macromolecules, destroy the interaction between molecules, form low molecular substances or polar groups with strong activity, make the fiber structure unstable or even damaged, cause fiber deformation and lead to performance failure. Therefore, the longer the sunshine time, the more the fiber is damaged, and the more microplastics are released during washing.

In terms of the size of the K value, the larger the K value, the more fabric microplastics are released. The optimization scheme of the factors is A3B3C3—that is, when the fabric structure is satin, the amount of friction is 10,000, and the sunshine time is 50 h, the release of fabric microplastics will reach the maximum. It can be seen from the data in Table 9 that the release of microplastics increases with the change of fabric structure from plain to twill to satin, the increase of the amount of friction and sunshine time. It shows that the release of microplastic is directly proportional to the number of weaving points of woven fabric and also to the amount of friction and sunshine time. Further analysis of variance is conducted for this orthogonal test, and the analysis results are shown in Table 10.

According to the ANOVA results in Table 10, the amount of friction has a very significant effect on the release amount of microplastics, while the fabric structure and sunshine time have no significant effect. Among the three factors, the F-value of the amount of friction is the largest, and the *p*-value is the smallest. Friction between fabric and other substances inevitably occurs in daily use, which leads to wear and the destruction of the fiber surface in the fabric. The larger the friction, the greater the degree of wear and destruction, the more fiber debris loss and fiber breakage, and the more microplastics released during washing. The fabric structure has no significant effect on the release amount of microplastics because the three kinds of woven fabrics are all compact structures. Although the release amount of microplastics is different for different fabric tissues due to the different number of interleaving points, the difference is not large enough. The reason for the insignificant effect of sunshine time on the release amount of microplastics is that the sunshine time selected in the experiment is relatively short. Although the release amount of microplastics increases with the increase of sunshine time, the difference in the release amount of microplastics under different sunshine times is not large enough.

Using mathematical software MATLAB to draw the three-dimensional surface map and contour map and to analyze the relationship between various factors and the amount of microplastic shedding, as shown in Figure 7.

According to Figure 7, under the same amount of friction, the release amount of textile microplastics of the satin fabric is the largest, followed by twill fabric, and plain fabric is the least. The release amount of textile microplastics increased slightly with the increase in sunshine duration, but the effect is insignificant. With the same fabric structure, the more friction, the more microplastics are released; The release amount of microplastics increased slightly with the increase of insolation time, but the effect was insignificant. Under the same exposure time, the release amount of textile microplastics of satin fabrics is the largest, followed by twill fabrics, and plain fabrics is the least. The more amount of friction, the greater the release of textile microplastics.

### 3.3. Morphology and Quantity of Microplastics

#### 3.3.1. Morphology of Microplastics

The filtered microporous filter membrane is observed using an HD002C-type fiber fineness analyzer, and the images of fibers released from woven and knitted fabrics are collected, as shown in Figure 8 and Figure 9.

From the observations in Figure 8 and Figure 9, it can be seen that the microfibers shedded by woven fabrics are slightly radian and mostly flat, while those fibers shedded by knitted fabrics are generally curved. The reason is the coil structure of the knitted fabric. The coil structure compresses the yarn, bends the yarn, and then further bends the fiber, forming a plastic deformation over time. After washing, the microfibers tend to escape the fabric at the coil length, which is significantly different from the woven fabric.

#### 3.3.2. Quantity of Microplastics

An HD002C fiber fineness analyzer was used to determine the average diameter Di (μm) and the average length Li (μm) of the microplastics. Referring to the data, the density ρ of polyester was 1.38 g/cm^3^, the average weight of the microplastics was set as Mi, and the number of microplastic roots was set as A.
(1)$$A\;=\frac{4\mathrm{Mi}}{\rho \pi D^{2}\mathrm{Li}}$$

The average diameter and length of the microplastics dropped from the sample are shown in Table 11.

The date is then placed into Equation (1) to obtain the number of microplastic shedded from each sample, as shown in Table 12.

As can be seen from Table 12, the release amount of microplastics in polyester knitted fabric is greater than that in polyester woven fabric. The reason is that woven fabrics have a greater degree of tension and tighter structure, while the coil string sleeve structure of knitted fabrics makes for a looser fabric structure, and the release of microplastics is easier to shed during washing. Among the three basic fabrics, satin fabrics had the largest release of textile microplastics, followed by twill fabrics, and plain fabrics had the smallest release of textile microplastics. The reason is that satin fabrics have the least interleaving points, and twill fabrics have more interleaving points. The most interleaving points of plain fabric make for the tightest degree of the three fabrics in the order of plain > twill > satin. The tightness of the fabric is the direct cause of the removal of microplastics during washing.

## 4. Conclusions

This paper studies the microfiber shedding of textiles in daily life through the simulation of household washing tests to explore the relationship between textile microplastics and fabric structure; the main conclusions are as follows.

A. Under the same washing conditions, the amount of microplastics released by knitted fabrics is greater than that of woven fabrics. The best washing scheme included a washing time of 80 min, a washing temperature of 50 °C, and a number of steel balls in the container of 30. According to the analysis of variance, the influences of fabric type, washing time and the number of steel balls on the release amount of microplastics are significant. The fabric type has the most significant effect on the release amount of microplastics. The change in washing temperature has almost no effect on the amount of microplastics released.

B. The release amount of textile microplastics from satin fabric is the largest, followed by twill fabric, and the release amount of textile microplastics from plain fabric is the smallest under the same washing conditions. The variance analysis showed that the friction times significantly affect textile microplastic shedding, but the fabric structure and sunshine time have no significant effect. With the increase in the amount of friction, the release of microplastics increases significantly. With the increase in sunshine time, the release of microplastics in textiles increases slightly, but the effect is insignificant.

It can be seen from the above conclusions that the relationship between the shedding of textile microplastics and fabric structure is primarily related to the weaving method. Knitted fabrics shed more microplastics than woven fabrics. The reason is that the knitted fabric’s coil structure gives the fabric structure higher scalability, softness and porosity, and the fiber is easier to pull out from the yarn and fall off. The woven fabric is composed of the warp yarns arranged at a certain angle, and the weft yarns arranged vertically to form a stable interweave structure. The structure is tight compared with the knitted fabric; the fiber is difficult to fall off during washing. The second is the relationship with the specific type of woven fabric interweave structure. The amount of microplastic shedding is in the order of satin > twill > plain. The reason is that the plain fabric has the most interweaving points and the tightest structure, and the twill fabric has the following interweaving points and the tighter structure. Of the three types of fabric structure, the satin fabric has the least interweaving points and the loosest structure, so the fibers in the loose structure are more likely to fall off the fabric’s surface during washing. In conclusion, there is a close relationship between the amount of textile microplastic shedding and the fabric structure. Therefore, when it comes to fabric design especially related to environmental protection and pollution, we should consider reducing the amount of textile microplastic shedding as much as possible from the perspective of fabric structure design to protect the environment.

## Figures and Tables

**Figure 1 polymers-14-05309-f001:**
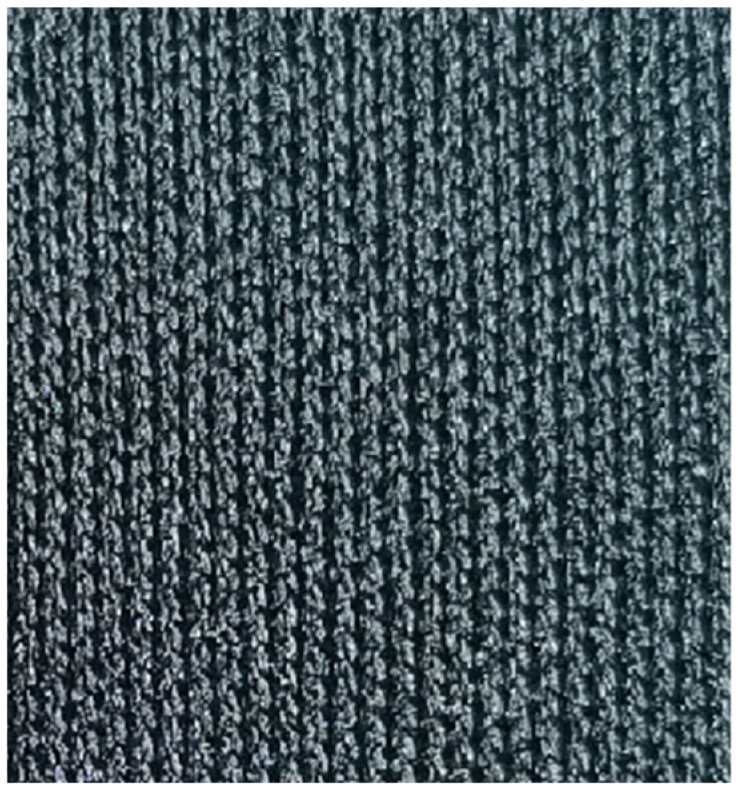
All-polyester knitted fabric.

**Figure 2 polymers-14-05309-f002:**
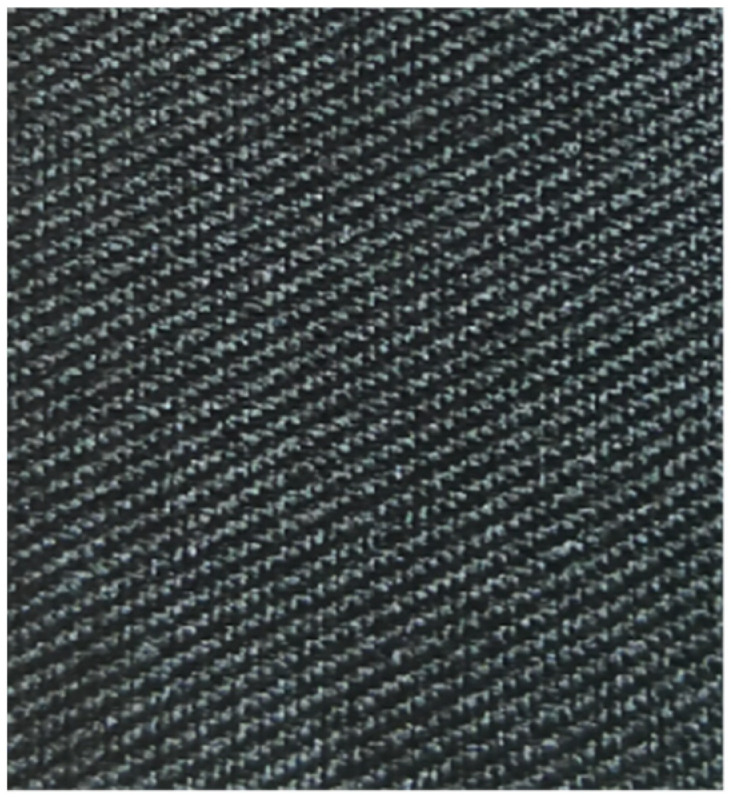
All-polyester woven fabric.

**Figure 3 polymers-14-05309-f003:**
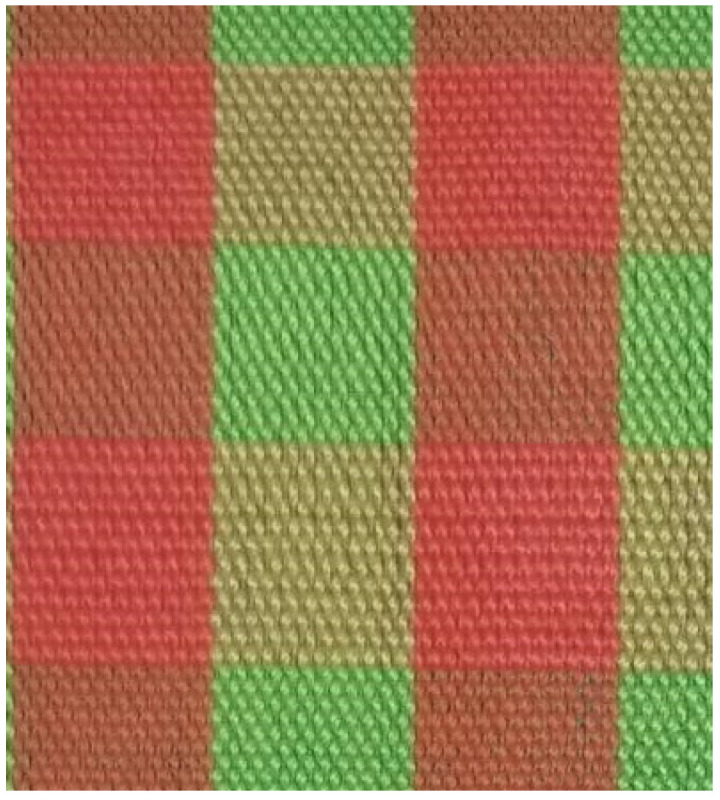
Plain fabric.

**Figure 4 polymers-14-05309-f004:**
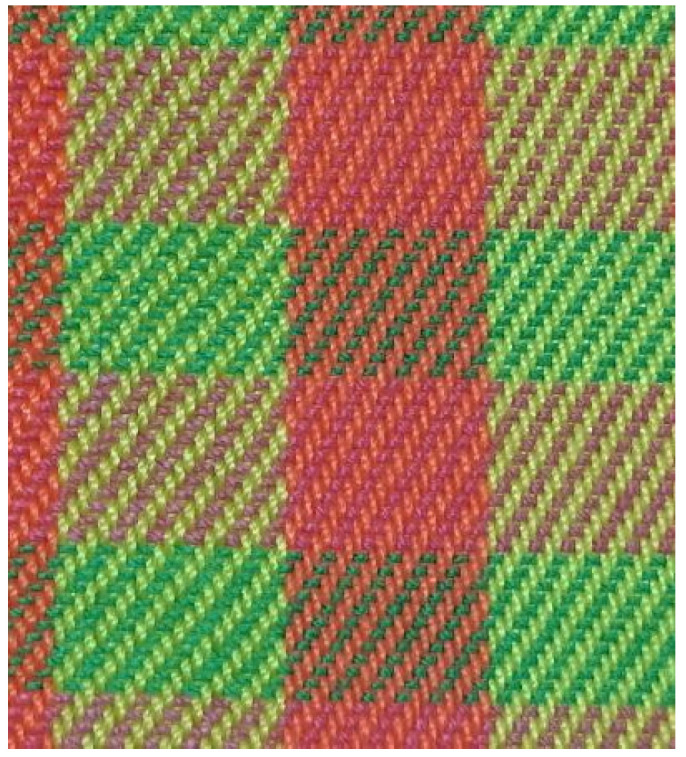
Twill fabric.

**Figure 5 polymers-14-05309-f005:**
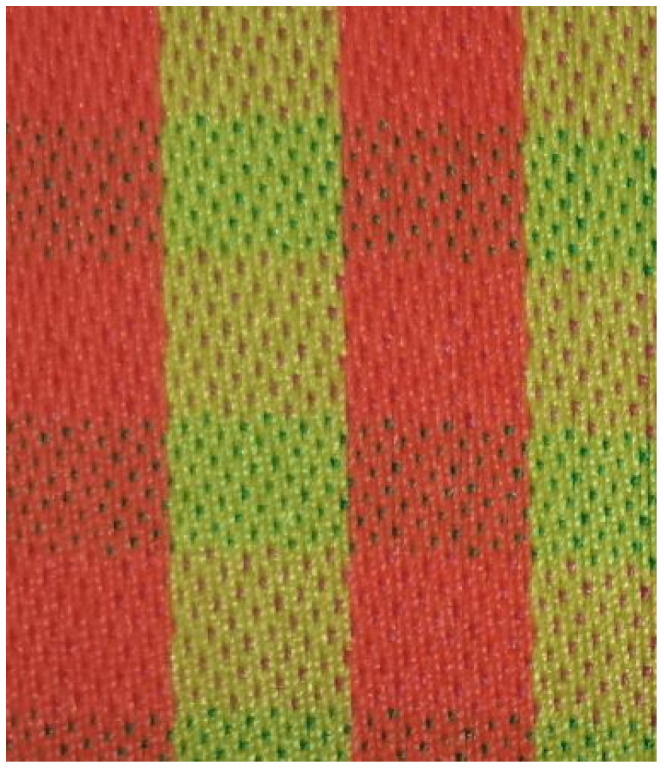
Satin fabric.

**Figure 6 polymers-14-05309-f006:**
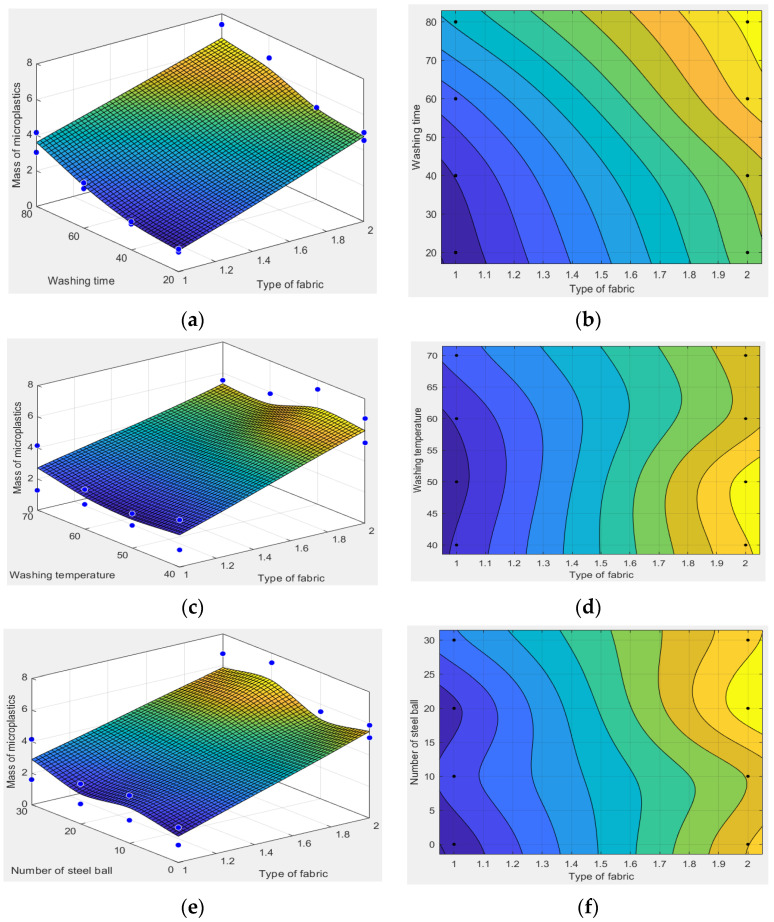
(**a**) Three-dimensional surface diagram of fabric type and washing time; (**b**) contour plot of fabric type and washing time; (**c**) three-dimensional surface diagram of fabric type and washing temperature; (**d**) contour plot of fabric type and washing temperature; (**e**) three-dimensional surface diagram of fabric type and steel ball count; (**f**) contour plot of fabric type and steel ball count. (The blue dots and black dots in the figure represent the positions of the values of the 16 experimental points respectively).

**Figure 7 polymers-14-05309-f007:**
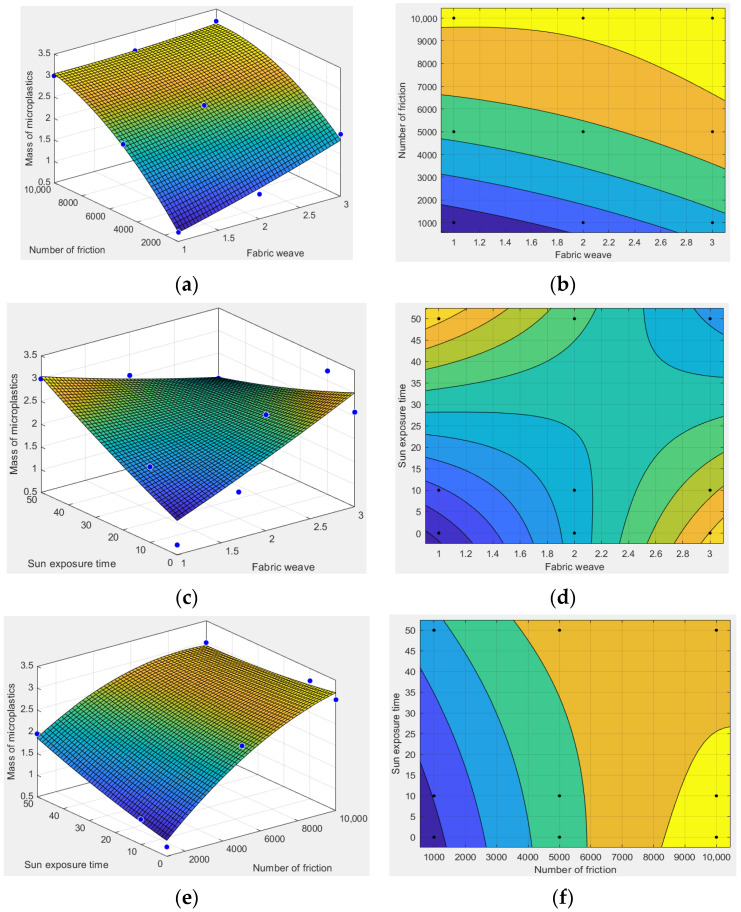
(**a**) Three-dimensional surface diagram of fabric structure and amount of friction; (**b**) contour plot of fabric structure and amount of friction; (**c**) three-dimensional surface diagram of fabric structure and insolation time; (**d**) three-dimensional surface diagram of fabric structure and insolation time; (**e**) three-dimensional surface diagram of the amount of friction and sunshine duration; (**f**) contour plot of the amount of friction and sunshine duration. (The blue dots and black dots in the figure represent the positions of the values of the 9 experimental points respectively).

**Figure 8 polymers-14-05309-f008:**
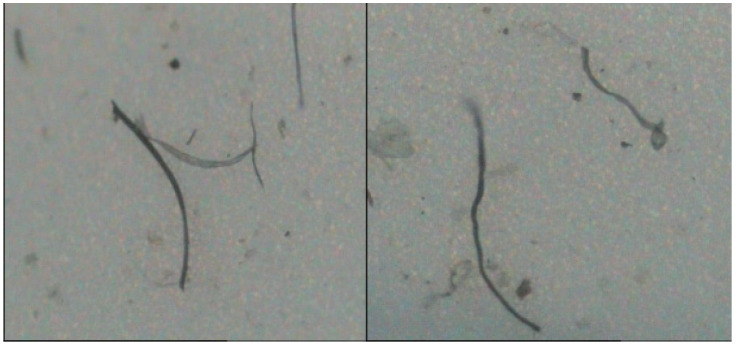
Fibers released from woven fabrics.

**Figure 9 polymers-14-05309-f009:**
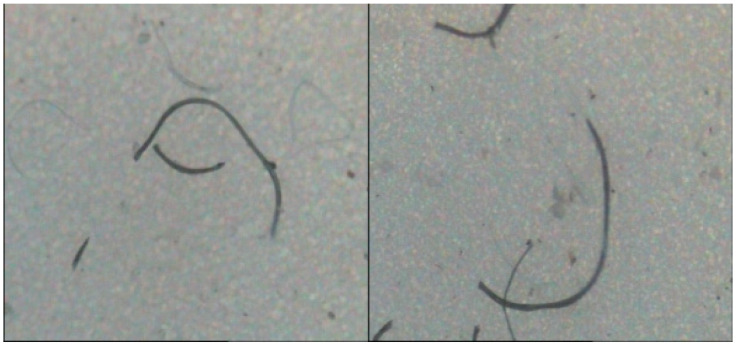
Fibers released from knitted fabrics.

**Table 1 polymers-14-05309-t001:** Characteristics of the tested fabrics.

Material	Group	Naming	Type of Fabric	Thickness [mm]	Area [cm^2^]	Fabric Structure	Warp Yarn Count (Ne)	Weft Yarn Count (Ne)	Coil Count (Ne)	Manufacturer
100% Polyester	A	A1	Knitted fabric	0.42	20 × 20	plain			20	Boran Textile
		A2	Woven fabric	0.43	20 × 20	twill	30	20		Golden Phoenix Fabric
100% Polyester	B	B1	Woven fabric	0.90	152	plain	20	20		
		B2	Woven fabric	1.04	152	twill	20	20		
		B3	Woven fabric	1.20	152	satin	20	20		

**Table 2 polymers-14-05309-t002:** Test equipment.

Serial Number	Product Model	Name	Manufacturer
1	SE-12AⅡ	Colorfastness tester	Wenzhou Darong Instrument Co., Ltd.
2	HD002C	Fiber fineness tester	Nantong Hongda Experimental Instrument Co., Ltd.
3	YM-10	Diaphragm vacuum pump	Jianhu Yamai Glass Instrument Technology Co., Ltd.
4	101A-3	Digital display constant temperature drying oven	Changzhou Dahua Instrument Co., Ltd.
5	Medium B-LB01-M	Aluminum weighing plate	Changde Bikman Biotechnology Co., Ltd.
6	Mixed cellulose ester	Microporous membrane filter	Suzhou Baizhi Material Technology Co., Ltd.
7	AUY-200	Electronic balance	Shimadzu Manufacturing, Japan
8	YG502	Pilling tester	Laizhou Electronic Instrument Co., Ltd.

**Table 3 polymers-14-05309-t003:** Level of factor for fabric type and washing conditions.

Level	A (Type of Fabric)	B (Washing Time/min)	C (Washing Temperature/°C)	D (Number of Steel Balls)
1	Knitted fabric	20	40	0
2	Woven fabric	40	50	10
3		60	60	20
4		80	70	30

**Table 4 polymers-14-05309-t004:** Orthogonal experimental design.

Test Plan	A (Type of Fabric)	B (Washing Time/min)	C (Washing Temperature/°C)	D (Number of Steel Ball)	Vacant Column
1	1	1	1	1	1
2	1	2	2	2	2
3	1	3	3	3	3
4	1	4	4	4	4
5	1	1	4	3	2
6	1	2	3	4	1
7	1	3	2	1	4
8	1	4	1	2	3
9	2	1	2	4	3
10	2	2	1	3	4
11	2	3	4	2	1
12	2	4	3	1	2
13	2	1	3	2	4
14	2	2	4	1	3
15	2	3	1	4	2
16	2	4	2	3	1

**Table 5 polymers-14-05309-t005:** Level of factor for fabric structure and external conditions.

Level	A (Fabric Structure)	B (Friction Times)	C (Sunshine Duration/h)
1	plain	1000	0
2	twill	5000	10
3	satin	10,000	50

**Table 6 polymers-14-05309-t006:** Orthogonal experimental design.

Test Plan	A (Fabric Structure)	B (Friction Times)	C (Sunshine Duration)	Vacant Column
1	1	1	1	1
2	1	2	2	2
3	1	3	3	3
4	2	1	2	3
5	2	2	3	1
6	2	3	1	2
7	3	1	3	2
8	3	2	1	3
9	3	3	2	1

**Table 7 polymers-14-05309-t007:** Test results and visual analysis.

Test Plan	A (Type of Fabric)	B (Washing Time)	C (Washing Temperature)	D (Number of Steel Balls)	Vacant Column	Microplastics Mass (mg/10 g)
1	1	1	1	1	1	1.11
2	1	2	2	2	2	1.46
3	1	3	3	3	3	2.54
4	1	4	4	4	4	4.14
5	1	1	4	3	2	1.27
6	1	2	3	4	1	1.59
7	1	3	2	1	4	2.22
8	1	4	1	2	3	3.03
9	2	1	2	4	3	4.99
10	2	2	1	3	4	5.17
11	2	3	4	2	1	5.52
12	2	4	3	1	2	5.88
13	2	1	3	2	4	4.55
14	2	2	4	1	3	5.09
15	2	3	1	4	2	6.73
16	2	4	2	3	1	7.38
K1	17.36	11.92	16.04	14.30	15.60	
K2	45.31	13.31	16.05	14.56	15.34	
K3		17.01	14.56	16.36	15.65	
K4		20.43	16.02	17.45	16.08	
k1	2.17	2.98	4.01	3.575	3.90	
k2	5.66	3.33	4.13	3.64	3.84	
k3		4.25	3.64	4.09	3.91	
k4		5.11	4.01	4.36	4.02	
R	3.49	2.13	0.49	0.79	0.19	

**Table 8 polymers-14-05309-t008:** Variance analysis of fabric type and washing conditions.

Sources of Variation	Bias Squares	Degree of Freedom	Mean Square	F	*p*	Significance
A (Type of Fabric)	48.8251	1	48.8251	2077.6638	0.00	**
B (Washing time)	11.0213	3	3.6738	156.3319	0.00	**
C (Washing temperature)	0.4089	3	0.1363	5.800	0.09	
D (Number of steel balls)	1.6883	3	0.5628	23.9489	0.01	**
E (Error)	0.0705	3	0.0235			
Sum	62.0141					

“**” indicates that this factor significantly impacts the amount of microplastics released.

**Table 9 polymers-14-05309-t009:** Test results and visual analysis.

Test Plan	A (Fabric Structure)	B (Number of Friction)	C (Sunshine Duration)	Vacant Column	Microplastics Mass (mg/10 g)
1	1	1	1	1	0.72
2	1	2	2	2	2.16
3	1	3	3	3	2.99
4	2	1	2	3	1.08
5	2	2	3	1	2.54
6	2	3	1	2	3.05
7	3	1	3	2	1.95
8	3	2	1	3	2.58
9	3	3	2	1	3.21
K1	5.87	3.75	6.35	6.47	
K2	6.67	7.28	6.45	7.16	
K3	7.74	9.25	7.48	6.65	
k1	1.96	1.25	2.12	2.16	
k2	2.22	2.43	2.15	2.39	
k3	2.58	3.08	2.49	2.22	
R	0.62	1.83	0.38	0.23	

**Table 10 polymers-14-05309-t010:** Variance analysis of fabric structure and external conditions.

Sources of Variation	Bias Squares	Degree of Freedom	Mean Square	F	*p*	Significance
A (Fabric structure)	0.5869	2	0.2935	13.7447	0.07	
B (Amount of friction)	5.1769	2	2.5885	121.2389	0.0082	**
C (Sunshine duration)	0.2609	2	0.1305	6.1101	0.14	
D (Error)	0.0854	2	0.0427			
Sum	6.1101					

“**” indicates that this factor significantly impacts the amount of microplastics released.

**Table 11 polymers-14-05309-t011:** Average diameter and average length of microplastics.

Research Materials	Average Diameter (μm)	Average Length (μm)
All-polyester woven fabric	16.3	374.8
All-polyester knitted fabric	18.1	518.3
Three basic fabrics	14.6	637.1

**Table 12 polymers-14-05309-t012:** Quantity of microplastics.

Fabric Type	Mass of Microplastics (mg/10 g)	The Number of Microplastics/10 g
All-polyester woven fabric	2.17	20,100 ± 20
All-polyester knitted fabric	5.66	28,200 ± 70
plain	1.96	13,300 ± 20
twill	2.22	15,000 ± 90
satin	2.58	17,500 ± 40

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
