# Peer review of "Study on the Relationship between Textile Microplastics Shedding and Fabric Structure"

_polymers, 2022, doi:10.3390/polym14235309_

Round 1

Reviewer 1 Report (Previous Reviewer 2)

it is fine

Reviewer 2 Report (Previous Reviewer 3)

Authors have addressed all my comments and I have no further question.

Reviewer 3 Report (Previous Reviewer 1)

Accept

Reviewer 4 Report (New Reviewer)

The current work studies the relationship between Textile Microplastics Shedding and Fabric Structure.

After a careful reading, I find the paper well written and well organized. I recommend the publication of the paper in its current form.

This manuscript is a resubmission of an earlier submission. The following is a list of the peer review reports and author responses from that submission.

Round 1

Reviewer 1 Report

1-    Please explain what is “vacant column” in the experimental designs.

2-    Explain about the K1-K4 in table 7.

3-    The results are not well presented and the discussions are weak.

4-    Statistical analyses are also weak.

Reviewer 2 Report

The manuscript entitled, study on the relationship between textile microplastics and fabric structure has been reviewed. The topic is very interesting and needed in today’s industry, but the work is not done in-depth enough to be considered fruitful to extract the results. The work done is not enough to be considered in such a highly reputed journal, i.e., Polymers. A more detailed analysis is needed. The work should be submitted to textiles-MDPI, a very emerging journal and more related to this research area.

There are several concerns related to the results; as mentioned in the abstract, plain woven fabrics had higher microplastic content than the satin ones; also, it is said that it was related to friction, then interwoven friction is more in plain woven fabrics. Can you please justify or elaborate on your results?

Provide more detailed materials; what was the fabric construction? What were yarn counts used?

Show your arrangement of tests. Consider adding schematics diagrams of the protocol.

I could not find any rationale for Figures 8 and 9. What do the authors want to show? There is no description added to the images. What is the difference between the two images added in these two Figures? Why one looks pink in color? It is not clear the need for Figures 8 and 9. What do the authors want to show from this figure?

Please add the name of manufacturers in Table 2, or better write them in wording. Also, mention under what standard they were used where applicable. Provide the instrument manufacturer, city, and country for all materials and equipment.

I could not find a logical reason for adding one set of knitted fabrics in this study.

There are several problems with the language of the manuscript. Therefore, the manuscript should be revised by a native speaker with good knowledge of the subject.

Add statistical error in Table 12.

Please check the typos, spelling errors, and unnecessary spaces carefully. For example, check the subscript used. Also, check the font consistency while writing; the authors might refer to the first paragraph on page 8.

References added are too few.

What are the prospects? Add a separate section or briefly add a few sentences in the conclusion section. Also, no research limitation is explained aligned with the research problem. The research limitations describe what dimensions of the problem are excluded by you and your study’s boundaries.

Reviewer 3 Report

Overall, this is an interesting study that illustrates the impact of washing conditions on the releasing of microplastics on different type of fabrics. I have some questions before publishing the manuscript.

The authors have mentioned ‘friction times’ multiple times. What is ‘friction time’? Since this is one of the important parameters of washing conditions, I strongly suggest authors clearly define this parameter before introducing the results.

What is a washing ball? Is it laundry ball? Or detergent pods? It would be better if authors provide an illustration on this as well.

Since friction is the mechanism that wears off the microplastics, I am wondering what type of lubricated friction it is. My guess would be elastohydrodynamic lubrication since it’s the friction on the polymer fabric, which is the soft material. If so, authors should cite some papers on elastohydrodynamic lubrication. For example:

Peng, Yunhu, et al. "Elastohydrodynamic friction of robotic and human fingers on soft micropatterned substrates." Nature Materials 20.12 (2021): 1707-1711.